behaviour, ecology, environmental science

ALAN, environmental change, Lampyridae, mate choice, sexual selection, signal

**Author for correspondence:**
Christina Elgert
e-mail: christina.elgert@helsinki.fi

# Reproduction under light pollution: maladaptive response to spatial variation in artificial light in a glow-worm

Christina Elgert[1,3], Juhani Hopkins[2,3], Arja Kaitala[2,3] and Ulrika Candolin[1,3]

[1]Organismal and Evolutionary Biology, University of Helsinki, PO Box 65, 00014 Helsinki, Finland
[2]Department of Ecology and Genetics, University of Oulu, PO Box 3000, 90014 Oulu, Finland
[3]Tvärminne Zoological Station, University of Helsinki, J.A. Palméns väg 260, 10900 Hanko, Finland

CE, 0000-0002-4609-5223; JH, 0000-0003-4724-2755; UC, 0000-0001-8736-7793

The amount of artificial light at night is growing worldwide, impacting the behaviour of nocturnal organisms. Yet, we know little about the consequences of these behavioural responses for individual fitness and population viability. We investigated if females of the common glow-worm *Lampyris noctiluca*—which glow in the night to attract males—mitigate negative effects of artificial light on mate attraction by adjusting the timing and location of glowing to spatial variation in light conditions. We found females do not move away from light when exposed to a gradient of artificial light, but delay or even refrain from glowing. Further, we demonstrate that this response is maladaptive, as our field study showed that staying still when exposed to artificial light from a simulated streetlight decreases mate attraction success, while moving only a short distance from the light source can markedly improve mate attraction. These results indicate that glow-worms are unable to respond to spatial variation in artificial light, which may be a factor in their global decline. Consequently, our results support the hypothesis that animals often lack adaptive behavioural responses to anthropogenic environmental changes and underlines the importance of considering behavioural responses when investigating the effects of human activities on wildlife.

## 1. Introduction

Artificial light is a growing global problem that changes the natural cycles of light under which species have evolved, such as daily, lunar, and seasonal cycles [1–3]. Vast areas of the Earth are currently lit up at night by direct light emission from various sources, such as streetlights, advertisements, buildings, and cars, and by indirect skyglow [4,5]. Artificial light is usually much brighter than natural night light and consequently likely to induce maladaptive changes in behaviours adapted to natural light cycles, such as trophic interactions and reproductive activities, and thereby reduce the fitness of individuals. Reduced fitness could, in turn, have negative consequences for the viability of populations and, ultimately, for ecosystem structure and functioning [6–11].

An increasing number of studies demonstrate that species alter their behaviour in response to artificial light, such as their foraging [12], predator avoidance [13], orientation [14], and timing of reproductive activities [15], but whether the responses are adaptive or not is largely unknown. The impact of anthropogenically induced changes to behaviour depends on past conditions and the evolutionary history of the species; species that encounter novel conditions may not have evolved reaction norms for coping with the disturbances and, hence, may express maladaptive responses [10,11].

Organisms that are particularly likely to be affected by artificial light at night are those active at dusk or at night, and who use light signals in their communication [16,17]. Prominent examples of such organisms are glow-worms and fireflies (Lampyridae). These beetles emit bioluminescent signals to attract and evaluate mates, such as prolonged glows or flashes of light, usually in the evening and at night when natural environmental light levels are low [18,19]. Artificial light could influence these light signals by inducing individuals to alter their emissions or by hampering the ability of individuals to detect or evaluate the signals. Such changes could affect the mating success of individuals and thereby the number of offspring they produce or the viability of their offspring [20], hence influencing population dynamics [20,21]. Consequently, artificial light at night has been postulated to be one cause of the current global decline of glow-worms and fireflies [22–24].

Previous studies have investigated some effects of artificial light at night on the signalling systems of glow-worms and fireflies. Firebaugh & Haynes [25] found artificial light to reduce the flashing activity of the firefly *Photuris versicolor*, but not of *Photinus pyralis*, whereas Costin & Boulton [26] found artificial light to reduce the flashing activity of several *Photinus* species. Ineichen & Rüttimann [27] and Bird & Parker [28] found, in turn, street lighting to interfere with the ability of the common glow-worm *Lampyris noctiluca* to attract and locate mates. However, the factors behind this variation in vulnerability to artificial light, and the degree to which species can reduce negative effects on mating success, are unknown.

In the common glow-worm, sedentary females emit a constant glow during a few nights to attract flying males [18]. The glow is produced in the lantern on the underside of the sixth and seventh abdominal segments through chemical reactions [29], and the intensity of the glow correlates positively with success in mate attraction [18,29]. Light conditions vary in nature depending on the phase of the moon and cloud conditions, therefore, glow-worms could have evolved a signalling system that maximizes signal transmission under variable light conditions [30,31]. For example, females may search out more shaded locations under moonlit nights, or delay the initiation of glowing if darkness falls later. Such responses could depend on the signal intensity of the female, as more intense signals should be easier to discern because of their higher contrast to the background. However, evolved responses may not be adaptive under artificial light, as the light is often brighter and more constant than natural light at night.

Glow-worms are capital breeders that depend on energy accumulated as larvae (they do not feed as adults). Consequently, there are energy restrictions on the number of nights they can glow as well as on egg laying [18]. If female responses to artificial light at night delays mating, females may have fewer energy reserves to invest in eggs or run the risk of remaining unmated, decreasing lifetime fecundity [32]. Thus, some behavioural responses to artificial light could be maladaptive and decrease offspring production, which could be one cause of their global decline [22–24]. To evaluate the degree to which light pollution could have contributed to their decline, more information is needed on the response of glow-worms to artificial light and how the response influences fitness components such as mating success.

We investigated if female glow-worms can mitigate negative effects of artificial light on mate attraction by adjusting the timing and location of glowing to spatial variation in artificial light, and whether the responses depend on their glow intensity. To investigate if females adjust their timing and location of glowing to artificial light, we exposed females to a gradient of artificial light in the laboratory and recorded effects on time glowing and movement in relation to the light source. We explored whether the responses depend on their glow intensity by comparing females with different glow intensity (measured as pronotum width [29]). Finally, we investigated if the responses influence mate attraction, by recording the ability of dummy females with different glow intensity to attract males when exposed to varying intensities of artificial light from a simulated streetlight.

## 2. Material and methods

### (a) Behavioural responses to light pollution

We collected glow-worm females in June 2017 from the surroundings of Tvärminne Zoological Station (N 59°51′, E 23°14′) in Southern Finland. We collected them by hand at night and transported them to the laboratory, where we placed them in individual vials (diameter: 8 cm) containing fresh moss and leaves. The vials were kept at room temperature, approximately 21°C, at a 20 h/4 h light/dark cycle, as darkness is restricted to only 4 h during the height of the breeding season at this particular latitude.

We investigated the responses of females to artificial light on the night after capture. We used a 100 cm × 15 cm arena with a white light-emitting diode, LED light (5 mm, cold white; peak intensity approximately 0.32 µW nm$^{-1}$ (microwatts/nanometre) at 660 nm (red) with a secondary peak intensity of approximately 0.26 µW nm$^{-1}$ at 440 nm (blue), as measured with a spectrophotometer and integrating sphere), at one of the short ends (figure 1). Light intensity was 40 lx (lumen per square metre) at the lit end, 1.5 lx in the middle, and 0.5 lx at the dark end of the arena, measured with an AIRAM UVM-8 lx-metre. We covered the bottom of the arena with soil and placed a cardboard rectangle, 4 cm high, along the length of the arena (in the middle of it) for the female to perch on, as females usually climb up on suitable structures to enhance the visibility of the glow to flying males [28,33]. A seashell was placed at each short end of the arena for the female to hide under. We alternated the position of the LED light between the two short ends among replicates.

We had two treatments: a light treatment and a control. We started each replicate at 23.00 by placing a female on the cardboard in the middle of the arena and turning on the LED light in the light treatment while leaving the LED light unlit in the control. We recorded the position of the female every 20 min for 2 h and noted whether she had settled down and initiated glowing. The borders of the arena had markings 10 cm apart, which we used to determine the position of the female and calculate the distance moved during each 20 min period. We defined females as settled when they had kept the same position over at least two consecutive observations and recorded the location and time of settling. If a female settled multiple times, we used the last settling location and time in the analysis. We estimated distance moved by summing distances moved during each 20 min period. This could underestimate the distance moved, as females may not move in a straight line, however, this estimation was consistent across trials and treatments. To investigate whether glow intensity influences responses to artificial light, we measured the maximum width of the pronotum (the structure that covers the dorsal surface of

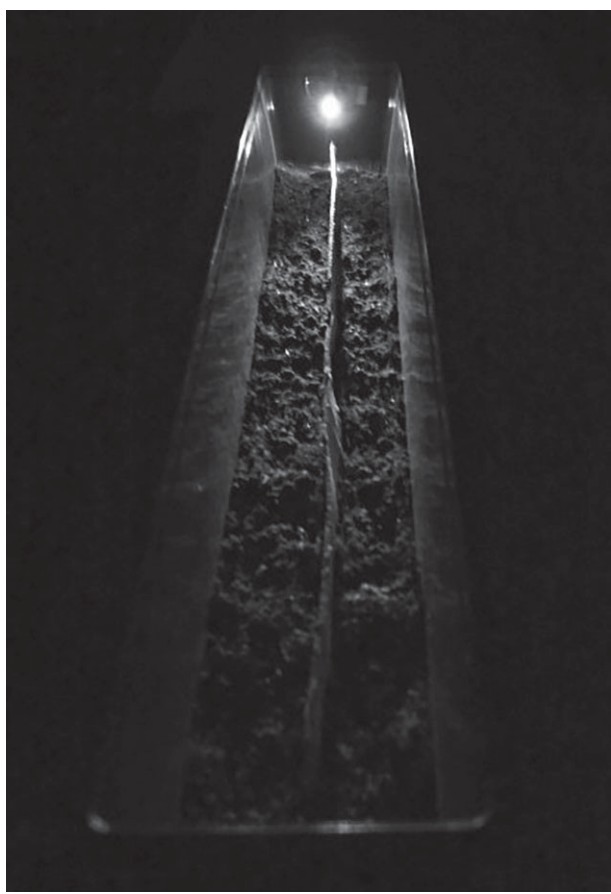

**Figure 1.** The experimental arena for investigating the effects of artificial light on the behaviour of female glow-worms: the timing of glowing and movement in relation to a light source. The arena had a white light-emitting diode, LED light, at one of the short ends, and a cardboard rectangle along the middle of the arena for perching.

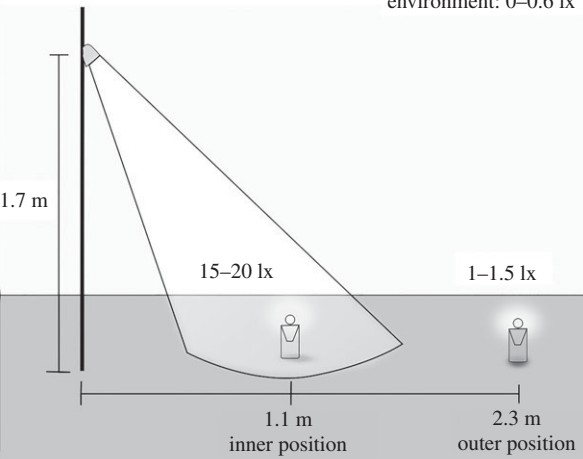

**Figure 2.** The design of the field experiment to investigate the impact of artificial light intensity on mate attraction. We placed two dummy females (constructed to trap males) at different distances from an artificial light source, a pole with a LED lamp: the female in the inner position was within the cone of light when lit and the other in the outer position outside the cone of light. We recorded the success of the two dummies in attracting males during the night.

the thorax) of the female, as this correlates with lantern size, which roughly correlates with glow intensity [29]. We tested 26 females in the light treatment and 63 females in the control.

To analyse the impact of the presence of artificial light and the pronotum width (glow intensity) of the female on behaviours that were binary, we used logistic regression, with separate models for each behaviour. The behaviours were: (i) whether a female started to glow, (ii) the direction of movement in relation to the light source (towards or away), (iii) whether a female hid under the shell or within the soil, and (iv) whether a female settled down.

To analyse the impact of artificial light on continuous measures, we used ANOVA, with separate models for each measure. The measures were: (i) total distance moved during the 2 h of observation and (ii) the settling location (for those that settled). Pronotum width (glow intensity) was included as a covariate in the initial models, but as the interaction terms and the main effects were non-significant, and the removal did not influence the significance of the other variables, it was removed in the final model. To analyse the influence of artificial light on the latency to glowing, we used a time-to-event analysis, Cox proportional hazards model [34]. The times were 'right-censored' for females that failed to glow by the end of the experiment. We checked that the data fulfilled the requirements of the analyses. All analyses were performed using SPSS 25. Significance was designated as $p < 0.05$.

## (b) Impact on mate attraction

We conducted the experiment from 6 June to 7 July 2017 in the surroundings of Tvärminne Zoological Station. We selected six sites that lacked shadowing trees or bushes, and where we had detected glow-worms during earlier years. We manipulated the presence of artificial light by erecting a pole and attaching a white LED light (ANSI FL1 Standard: 35 lumen, beam distance 24 m) at the height of 1.7 m (figure 2). The angle between the ground and the direction of the centre of the light was 55°.

We placed two dummy females (LED lures, see construction in [29]) at two different distances, in a straight line from the pole (figure 2). One dummy was placed in the inner position (within the cone of light from the pole when lit), at 1.1 m from the pole, at 15–20 lx (peak intensity approx. 0.08 μW cm$^{-2}$ nm$^{-1}$ at 455 nm, measured with a spectrophotometer and cosine corrector), which corresponds to light levels under common streetlights [2,35]. The other dummy was placed in the outer position, at 2.3 m from the pole, at 1–1.5 lx (peak intensity approx. 0.004 μW cm$^{-2}$ nm$^{-1}$ at 455 nm), which is slightly brighter than natural light levels at night in the area in the summer (0.1–0.6 lx, peak intensity 0.0003–0.0016 μW cm$^{-2}$ nm$^{-1}$ at 460 nm, measured on 31 May and 1 June 2020 at 12.30 in an open area on both an overcast and a moonlit night). The dummies were designed to trap males within a plastic bottle [29]. The wavelength of the LED lures was 562 nm, similar to female glow-worms (550–570 nm) [33,36]. To investigate the effect of glow intensity on mate attraction, we varied the glow intensity of the two dummies among replicates; peak glow intensity approximately 0.03, 0.06, and 0.13 μW nm$^{-1}$ (measured with a spectrophotometer and integrating sphere). The paired dummies within a replicate had the same glow intensity. The differences in glow intensity reflected natural variation in the wild ((A-M Borshagovski 2017–2018, unpublished data on spectrophotometer measurements).

We started each trial when dusk began to fall (approx. 22.00), by turning on the glow of the two dummy females, as well as the light from the pole in the artificial light treatment, while leaving poles unlit in the control. We checked the dummy females 3–4 h later (at 01.00–02.00) for the presence of males and turned off all lights. Males are unlikely to escape from the traps (C Elgert 2017–2020, personal observation).

We conducted 38 replicates of the artificial light treatment, with three dummy glow intensities (low: $n = 13$; medium: $n = 14$; high: $n = 11$, with two dummy females in each replicate) and 19 replicates of the control, with three dummy glow intensities (low: $n = 6$; medium: $n = 5$; high: $n = 8$, with two dummy females in each replicate). We distributed the treatments (presence of artificial light and glow intensities) equally among the six sites.

**Table 1.** Behavioural responses of glow-worm females exposed to a gradient of artificial light in an elongated arena. Logistic regression was used to analyse differences in proportions, and ANOVA to analyse differences in continuous variables.

| | light | control | | |
|---|---|---|---|---|
| **response** | **proportion** | **proportion** | **Wald $X_1^2$** | ***p*** |
| glowed | 6 out of 26 | 61 out of 63 | 28.29 | <0.001 |
| hid | 7 out of 26 | 1 out of 63 | 7.22 | 0.007 |
| moved away from light | 14 out of 26 | 30 out of 63 | 0.285 | 0.593 |
| settled down | 23 out of 26 | 54 out of 63 | 0.119 | 0.730 |
| | **mean ± s.e.** | **mean ± s.e.** | ***F*** | **d.f.** | ***p*** |
| distance moved (cm) | 63.58 ± 8.31 | 60.56 ± 5.56 | 0.088 | 1, 88 | 0.767 |
| distance settled (cm) | 3.48 ± 6.26 | −0.07 ± 3.55 | 0.274 | 1, 77 | 0.602 |

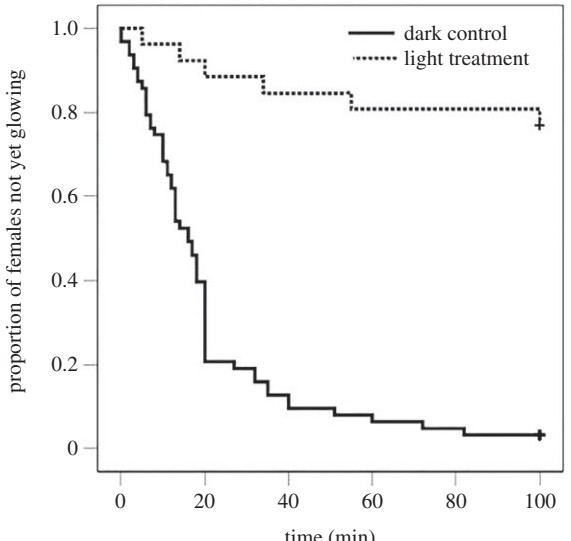

**Figure 3.** The latency until glowing of female glow-worms in the absence (control) and presence of artificial light. The graph shows Kaplan–Meier survival curves for latency. (+) indicates right-censored data.

In the analyses, we used the presence or absence of males in each dummy female trap as the response variable, rather than the number of males caught, to rule out the possibility that the presence of one male had attracted additional males. We analysed the data using a GLMM (generalized linear mixed model) with binomial error distribution and logit link function, with the presence or absence of male(s) in each dummy trap as the binary response variable. We used light treatment (light on or light off), position with respect to the light fixture (inner position or outer position), and glow intensity as fixed factors, and date and site as random factors, with site nested within date. We started with a full model and deleted non-significant interaction terms and fixed terms when this was supported by the Akaike information criterion (AIC) and did not reduce the significance of other terms [37]. All analyses were performed using SPSS 25. Significance was designated as $p < 0.05$.

# 3. Results

## (a) Behavioural responses to light pollution
Artificial light in the experimental arena reduced the probability that females glowed (table 1), increased the latency to glowing for those that glowed (Cox proportional hazard model, Wald = 30.599, $p < 0.001$, figure 3), and increased the probability that females went into hiding (table 1). Females that glowed continued to do so until the end of the trial, except for one female that glowed for 73 min under light and then ceased glowing. Artificial light did not influence the direction of movement, the distance moved, whether they settled down, or the distance from the light when they settled down (for those that settled) (table 1).

## (b) Impact on mate attraction
The effect of artificial light from a light pole on the probability that a dummy female attracted one or more males depended on the position relative to the light fixture; females in the inner position, inside the cone of light, were less successful in attracting males than females in the outer position (table 2 and figure 4). Mate attraction did not depend on the glow intensity of the dummy females ($F_{2,108} = 0.164$, $p = 0.85$, electronic supplementary material, figure S1), and the variable was removed from the final model as this did not influence the significance of the other variables.

# 4. Discussion
Our study is the first to demonstrate maladaptive behavioural responses to artificial light at night in female glow-worms; females are less likely to glow in the presence of artificial light and instead hide. They do not respond to spatial variation in light conditions by moving away from the light source, and if they glow under artificial light, they delay the onset. These responses appear maladaptive as females could improve their mating success by moving only a short distance from the light source. This was demonstrated in our field experiment that imitated conditions under a streetlight; success in mate attraction was significantly higher only 1.2 m further from the light source. Such small-scale spatial variation in light conditions is common in nature, in urban and rural housing areas, as well as along roads, which suggests that maladaptive responses to spatial variation in light conditions are common and reduce the mating success of glow-worms.

Our finding of low mate attraction success under artificial light aligns with earlier studies that found streetlights hamper mate attraction [27,28]. Females can respond to light, as they move away from green LED lights imitating

**Table 2.** The influence of the presence of artificial light, and the position relative to the light fixture—inner or outer—on the probability that a dummy female attracted one or more males. Analysis is based on GLMM with binomial error distribution. Random factors were date and site, with site nested within date.

| treatments | $F_{1,110}$ | $p$ |
|---|---|---|
| presence of light | 6.489 | 0.012 |
| position | 7.683 | 0.007 |
| presence × position | 5.767 | 0.018 |

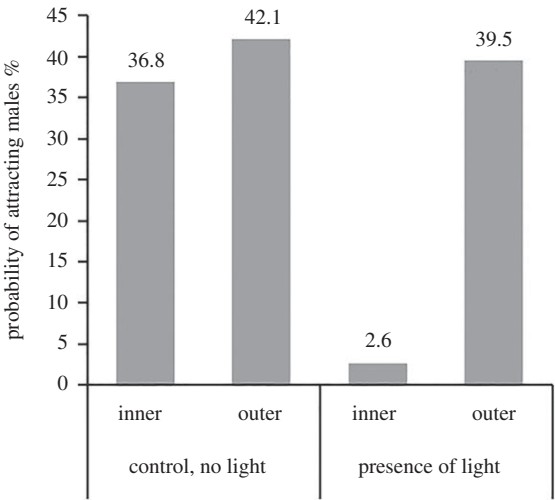

**Figure 4.** The probability that two dummy females attracted one or more males in the absence (control) and presence of artificial light from a light pole. In the presence of artificial light, one of the dummies was inside the cone of light and the other outside.

conspecific females [38], but bright artificial light from street-lights is stronger and of different wavelength than the glow of females, which may explain the different responses. During the day, females seldom move more than a few centimetres, as they spend the time in refuges [18,27,33,39], and relocation, therefore, depends on movement during the night. These are usually limited to short distances of 10–20 cm [18,19,39]. Thus, female glow-worms appear to lack adaptive reaction norms for responding to spatial variation in artificial light.

The lack of appropriate responses is probably a conse-quence of the natural absence of bright light at night. The selection has apparently not (yet) favoured the evolution of reaction norms for responding to bright light. Glow-worms respond to moonlight [30], but this is much weaker than the light reaching the ground from streetlights; according to Kyba et al. [40] typically 0.05–0.1 lx, versus 10–60 lx. Moreover, staying in one location under moonlight can be an adaptive strategy, as natural light conditions vary both during and between nights and moving uses up valuable energy, which can reduce fecundity, in these capital breeders [29,32,41].

Whether glow-worms could avoid illuminated locations before settling down and initiating the period of glowing is unknown. Earlier studies have found glowing females in illu-minated areas near streetlights [27], which indicates that females base their choice of display site on properties other than nightly light conditions (for example, habitat structure). Larvae have the potential to avoid light-polluted areas by moving towards more undisturbed environments before they pupate to adults, as they can move up to 5 metres per hour [18], and do not pupate before they are 2–3 years old [18,33,42]. However, whether larvae can detect and respond to light pollution is currently unknown. Importantly, large-scale avoidance of areas with artificial light would alter the distribution of glow-worms, which could accelerate their decline, as light pollution is increasing worldwide [24,35].

Artificial light at night has been postulated as a potential driver of evolutionary change [43]. However, whether glow-worms can evolve local adaptation to artificial light is unknown. Fireflies of North America differ in the wavelength of their glow depending on the local background, but the degree to which the differences have a genetic basis and indi-cate a potential for genetic adaptation is undetermined [44]. Evolutionary processes are generally slow, and likely to be particularly so in the glow-worm, which has a generation time of up to 3 years [19,28,42]. Moreover, the population could lack the genetic variation needed for evolutionary responses, or the responses could make the species less well adapted to natural light conditions, which would constrain the evolution of reaction norms suited to light-polluted conditions.

Interestingly, the response of females to artificial light did not depend on their supposed glow intensity, measured as pronotum width (which correlates with body and lantern size and, thus, with glow intensity [29]). Moreover, the glow intensities of the dummies did not influence mate attraction under artificial light, although the selected intensities reflected those of females in nature. These results suggest that the strong intensity of the artificial light overrode differences in the weaker glow intensity of females. In areas with low-intensity artificial light, differences among females in glow intensity may still influence mating success, as males are known to prefer brighter females [29,45,46]. A larger sample size may be needed to detect the weaker effect of glow inten-sity on mate attraction as no significant effect of glow intensity on mate attraction was detected in the controls.

In conclusion, our results support the suggestion that light pollution is one cause of the global decline of glow-worms and fireflies, together with other human-induced environmental changes, such as climate change, habitat destruction, and the spread of insecticides [19,22,47,48]. The recorded lack of adaptive responses decreases success in mate attraction, which could reduce the reproductive output of populations. This supports the hypothesis that many animals may lack adaptive responses for coping with anthropogenic environmental changes, in this case, because light conditions have been stable throughout their evolutionary past [10].

These results underline the importance of assessing the effects of artificial light on individual behaviour. Artificial light has been proposed as a driver of insect decline [22,23], and our results show how such an effect can arise through the lack of adaptive behavioural responses. Knowledge of the mechanisms behind the effects of artificial light on popu-lations is important, as it can improve our ability to develop environment-friendly lighting systems to minimize negative effects on wildlife [49]. The importance is increasing as con-ditions are expected to worsen with the expansion of the human population and the increased use of white LED lights [49,50]. At a broader level, our study illuminates the importance of investigating the fitness consequences of behavioural responses to human-induced environmental

changes. An increasing number of studies find organisms respond behaviourally to various human-caused disturbances, but whether the responses are adaptive or not, and how they impact fitness components, are less known [51,52]. This lack of knowledge hampers our ability to predict both short- and long-term effects of human disturbances on wildlife, as well as the development of effective management strategies to mitigate negative effects and requires more studies into the fitness consequences of behavioural responses to anthropogenic environmental changes.

Data accessibility. The raw data is available from the Dryad Digital Repository: https://doi.org/10.5061/dryad.2z34tmphw [53].

Authors' contributions. C.E., U.C., A.K., and J.H. designed the research, C.E. performed the experiments, C.E. and U.C. analysed the data, and C.E. and U.C. wrote the manuscript. All authors gave final approval for publication and agree to be held accountable for the work performed therein.

Competing interests. The authors declare no competing interests.

Funding. The work was funded by the Swedish Cultural Foundation in Finland (grant no. 148370 to C.E.), Maj and Tor Nessling Foundation (grant no. 202000239 to C.E.), and Academy of Finland (grant no. 294664 to A.K.).

Acknowledgements. We thank Anna-Maria Borshagovski and Gautier Baudry for assistance with the experiments, Topi Lehtonen and Lucy Katherine McLay for excellent comments on the manuscript, and Tvärminne Zoological Station for providing facilities.

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
