## [Reviewer comments · Proceedings of the Royal Society B: Biological Sciences]

Review History

RSPB-2019-2682.R0 (Original submission)

Review form: Reviewer 1 (Jana Eccard)

Recommendation

Major revision is needed (please make suggestions in comments)

Scientific importance: Is the manuscript an original and important contribution to its field?

Excellent

General interest: Is the paper of sufficient general interest?

Excellent

Quality of the paper: Is the overall quality of the paper suitable?

Acceptable

Is the length of the paper justified?

Yes

Should the paper be seen by a specialist statistical reviewer?

No

Do you have any concerns about statistical analyses in this paper? If so, please specify them explicitly in your report.

Yes

It is a condition of publication that authors make their supporting data, code and materials available - either as supplementary material or hosted in an external repository. Please rate, if applicable, the supporting data on the following criteria.

Is it accessible?

N/A

Is it clear?

Yes

Is it adequate?

Yes

Do you have any ethical concerns with this paper?

No

Comments to the Author

Review for RSPB-2019-2682, Jana Eccard, Animal Ecology, University of Potsdam

Paper: Reproduction under light pollution: maladaptive response to spatial variation in artificial light in a glow-worm

I love the experiment and the story, however I find the paper is not written very professional. Reading and commenting it felt like reading a student assignment, of a very good student but inexperienced paper writer (e.g. consistency in wording, finding functional terminology for the treatments, the function of a figure legend, the scope of the discussion). To improve this would have rather been the job of the senior authors before submission, not the job of a Proceedings reviewer. Still I kept reading and commenting below, since the experiments on glow worms are beautiful and light pollution is an important topic on the context of insect decline.

Abstract line 28: i do not understand this effect when reading the abstract only, please rephrase Intro

- I would exchange line 36/37 with line 38 and start the article with the adaptations of organisms to natural light cycles
- The intro on disturbing effects of ALAN could be extended to other signaling systems of animals with luminescence, e.g. in the sea
- Line 70 plural: lights

Methods

- Add information on scale and setup and purpose to the figure legend fig. 1
- wether, not if
- 105 body size
- the width of the pronotum of the female (the structure that covers the dorsal surface of the thorax) at its widest part – change to ... the maximum width of the pronotum (the structure that covers the dorsal surface of the thorax) of the female
- 107 measured it for what? ... as a proxy of female body size (19)
- Would latency to start glowing have been a variable to analyse?
- why are treatment and control so differently in numbers? Give a good reason since this is suspicious
- 109-116 state clearly at the start of the paragraph which are the predictor variables included to logistic regression
- 160 terms that were part of your set-up and hypothesis should not be removed from the model, even if non-significant (this is the statistic answer to your question!). You wanted to test

the importance of the Light intensity. It was not important but this is a result! Keep in in your models and discuss it in your discussion! (alternatively: show in the methods that it is not important, and shift the focus of the paper in hypotheses and analysis)

- Do not name one condition of the light treatment "light treatment", otherwise the reader cannot distinguish factor name and factor levels.
- Figure 2: state purpose of this set up (to attract glow worms). Figures with their legends should be extractable and useful outside the paper
- Complicated explanations in Lines 185-190 illustrate the problem with the name of the factor "distance", a confusing term for the location. Here it codes for a two-factor level categorical factor: illuminated or neighbouring, dark location. Meanwhile "distance" implies an interval-scaled gradient (something that increases or decreases with distance, make sure R really used as it "as factor"). In the methods you had named the same factor "locality", in Figure 2 distances are given in meters, in Figure 3 two levels "near" and "far". Be consistent. You should probably coin a better term, e.g. inside/outside light cone, factor "cone", or something similar, so everybody can understand your setup and analysis.
- Table 2: also provide information on the importance of your random factors date and locality (R marginal and R conditional)
- Fig. 3: the survival analysis <for proportion females not yet glowing> is somehow expressed around the corner. How about analyzing <nrs of females glowing>, or analyzing the <latency until female glows> (should yield similar results).
- Fig. 4: for a comprehensive legend you must give more information

The discussion is very limited. I miss considerations of

- predation risk for moving females at different light conditions (see e.g. Eccard et al. 2018 Ins Cons, Risk uniform and risk heterogenous conditions (Eccard and Liesenhohann 2008)
 - whether or not, or what kind of light heterogeneity occurs in nature
 - short term (this experiment) vs long term responses to illumination,
 - background knowledge on female and larval glow worm mobility - affecting their ability to leave or avoid lit areas as suggested
 - Sensory considerations, which light conditions allow male insects to pick up the signal, how sensory channels are affected by different ALAN techniques. Results of the mate attraction experiment should be picked up in the discussion
 - Importance of glow intensity of the dummy females: this variable was not significant but this needs to be discussed!!
- Line 241; direct
- Authors contribution statement in the beginning and end of the draft (line 252) are contradictory. Consider contribution of AK

Review form: Reviewer 2

Recommendation

Major revision is needed (please make suggestions in comments)

Scientific importance: Is the manuscript an original and important contribution to its field?

Good

General interest: Is the paper of sufficient general interest?

Good

Quality of the paper: Is the overall quality of the paper suitable?

Good

Is the length of the paper justified?

Yes

Should the paper be seen by a specialist statistical reviewer?

No

Do you have any concerns about statistical analyses in this paper? If so, please specify them explicitly in your report.

No

It is a condition of publication that authors make their supporting data, code and materials available - either as supplementary material or hosted in an external repository. Please rate, if applicable, the supporting data on the following criteria.

Is it accessible?

Yes

Is it clear?

Yes

Is it adequate?

Yes

Do you have any ethical concerns with this paper?

No

Comments to the Author

The authors present a study investigating the effects of direct light sources on mate attraction behaviors in the glow worm. They found that females under higher ALAN have decreased glowing onset or don't glow, that the females don't move to get away from the light pollution, and that males are less attracted to glowing females that are under direct light sources. The study is well done and could be published as is at a more specific journal, however, I feel that this study does not meet the high standards of novelty for Proc B as is currently written. However, I do think that the authors could add to this manuscript and make it very attractive to readers of Proc B.

As I was reading through the manuscript I was very curious as to what the difference in achromatic contrast was between the female glow and the ambient lighting. This would not be hard to measure and then model using a non-specific visual system and it would greatly add to the story. I suggest measuring the radiance of the glow worm lures at the angle that a male would be seeing them and also measuring the radiance of the background substrate (street, grass, etc?) that a male would be viewing the glow against. Do this for each of the different distances of the glow worm to see what the contrast is. This would predict if males can even see the glow of the glow worm under the different light treatments. If they easily can, then one could discuss how it isn't a visual detection issue but instead is likely a photophobic situation for males. I am very excited to see what you find.

Another avenue you could take it to set this paper apart from the previous work on Lampyridae, would be to look into the mechanism of the visual abilities of the female. What are they using to detect light levels to determine glowing or not? Are they using compound eyes or ocelli? If they are using only ocelli then it makes sense that they would not be able to determine where to move to for better lighting conditions. If they are using compound eyes, then what is their sensitivity to light levels and what is their acuity? Would they be able to see that there is an actual light gradient? I believe the authors are giving these beetles a lot of visual credit for being able to detect a light gradient of only one order of magnitude. This would be a very informative angle as well.

Lastly, and I would not suggest this as this would be the most time consuming, the authors could investigate actual fitness of these glowworms looking at mating rates and/or offspring production.

I hope my above suggestions are not overly daunting, but I do believe that this paper could take behavioral studies of light pollution to the next level and it is needed now in the field.

One minor detail that needs to be addressed - you use lux for ambient lighting but you use uW for radiance of glow worm lures. You need to either stick with photometric units (lux and lumens) or radiometric units (uWs or better yet, photons). This is very important.

Review form: Reviewer 3

Recommendation

Reject - article is not of sufficient interest (we will consider a transfer to another journal)

Scientific importance: Is the manuscript an original and important contribution to its field?

Acceptable

General interest: Is the paper of sufficient general interest?

Acceptable

Quality of the paper: Is the overall quality of the paper suitable?

Acceptable

Is the length of the paper justified?

Yes

Should the paper be seen by a specialist statistical reviewer?

No

Do you have any concerns about statistical analyses in this paper? If so, please specify them explicitly in your report.

No

It is a condition of publication that authors make their supporting data, code and materials available - either as supplementary material or hosted in an external repository. Please rate, if applicable, the supporting data on the following criteria.

Is it accessible?

N/A

Is it clear?

N/A

Is it adequate?

N/A

Do you have any ethical concerns with this paper?

No

Comments to the Author

The authors examine the movement patterns of female glowworms exposed to light pollution in experimental arenas in a laboratory. They also examine how light pollution impacts the likelihood that a female glowworm will mate under artificial lights in the field. The combination of complimentary field and laboratory experiments is one of this paper's assets.

This paper focuses on a very specific study organism--- glowworms. Many readers may find these organisms to be inherently interesting (insects that glow!). The addition of a few paragraphs in the introduction and discussion could give readers a broader perspective on how this study fits in with other research on how anthropogenic disturbances such as light pollution impact animal mating, movement, and courtship behaviors.

As I read through the paper, there were a few instances where I found myself wishing for more information. A few terms, such as "capital breeder" and "lantern," should be defined within the text for the benefit of readers who are unfamiliar with this study system. Further, the addition of more information about glowworm biology and natural history could help readers contextualize and interpret the results of this study. For example, readers may wish to know how far female glowworms typically travel at night, if this information is known.

Additional comments:

Line 30,68- The term "capital breeders" is used throughout this manuscript (e.g., lines 30, 68), but is not defined. This term may be unfamiliar to readers.

Line 37- Spell out "advertisements" instead of "adverts."

Line 89- What does the character immediately preceding "8 cm" in "(Ø 8 cm)" mean? Does this refer to one of the dimensions of the vials?

Line 93- Are light emission spectra available for the LED lights used in this experiment?

Line 95- Interesting! I'm trying to picture what the arenas might have looked like. Were the females most immediately in contact with the soil with the cardboard?

Line 99- Capitalize "pm."

Line 101- I'd like to know more about how the position of the female was recorded. Was the females' location marked within the area? Was the distance from the last known location measured with a ruler?

Line 103- "If a female settled multiple times, we used the last one as the settling location and time." How was "settling" defined? Additionally, I wonder if this approach would tend to underestimate the total distance moved by the female? For example, if a female moved 2 cm to the left, then 3 cm to the right along a straight line, would that be recorded as 5 cm or 1 cm traveled?

Line 105- Please define "lantern." Additionally, are there published data on the relationship between lantern size and female size? If so, please include a citation immediately after this statement: "To investigate if female size, which roughly correlates with lantern size and glow intensity,"

Line 123- Citation for this statistical analysis.

Line 148- How long do males stay with females after initial contact? Is it possible that some males might have left females prior to 1-2 AM, or is it more likely that males would stay with females

all night?

Line 148- Capitalize AM.

Lines 205, 209- What is known about the distances female glowworms typically travel in the field at night? How does 1.2 m relate to the distances females typically travel?

Decision letter (RSPB-2019-2682.R0)

15-Jan-2020

Dear Mrs Elgert:

I am writing to inform you that your manuscript RSPB-2019-2682 entitled "Reproduction under light pollution: maladaptive response to spatial variation in artificial light in a glow-worm" has, in its current form, been rejected for publication in Proceedings B.

This action has been taken on the advice of referees, who have recommended that very substantial revisions are necessary. With this in mind we would be happy to consider a resubmission, provided the comments of the referees are fully addressed. However please note that this is not a provisional acceptance.

Sincerely,
Professor Hans Heesterbeek
<mailto:proceedingsb@royalsociety.org>

Associate Editor
Board Member: 1
Comments to Author:
Dear Authors,

Your manuscript has now been assessed by three excellent reviewers in the field. Although all

reviewers see value in the topic they have valid and substantial concerns with the current version of the manuscript. These need to be specifically addressed if this paper is to compete for space in Proceedings B.

Namely, two reviewers mention issues with the language, description, and scope of the paper as written. This could be part of the reason why the strength in experimental design or findings of the paper are getting lost behind what could be vast improved writing.

Additionally, given that Proceedings is a general topic journal the authors must bring their relevance and findings more clearly into their broader implications in both the Abstract and Discussion; at least better than is done here.

It is my hope that these comments will help the authors in revising their manuscript for publication here. However, if the authors cannot address these issues and the others raised I think this manuscript would be best served somewhere else.

Reviewer(s)' Comments to Author:

Referee: 1

Comments to the Author(s)

Review for RSPB-2019-2682, Jana Eccard, Animal Ecology, University of Potsdam

Paper: Reproduction under light pollution: maladaptive response to spatial variation in artificial light in a glow-worm

I love the experiment and the story, however I find the paper is not written very professional. Reading and commenting it felt like reading a student assignment, of a very good student but inexperienced paper writer (e.g. consistency in wording, finding functional terminology for the treatments, the function of a figure legend, the scope of the discussion). To improve this would have rather been the job of the senior authors before submission, not the job of a Proceedings reviewer. Still I kept reading and commenting below, since the experiments on glow worms are beautiful and light pollution is an important topic on the context of insect decline.

Abstract line 28: i do not understand this effect when reading the abstract only, please rephrase Intro

- I would exchange line 36/37 with line 38 and start the article with the adaptations of organisms to natural light cycles
- The intro on disturbing effects of ALAN could be extended to other signaling systems of animals with luminescence, e.g. in the sea
- Line 70 plural: lights

Methods

- Add information on scale and setup and purpose to the figure legend fig. 1
- 105 wether, not if
- 105 body size
- 107 the width of the pronotum of the female (the structure that covers the dorsal surface of the thorax) at its widest part – change to ... the maximum width of the pronotum (the structure that covers the dorsal surface of the thorax) of the female
- 107 measured it for what? ... as a proxy of female body size (19)
- Would latency to start glowing have been a variable to analyse?
- 108 why are treatment and control so differently in numbers? Give a good reason since this is suspicious
- 109-116 state clearly at the start of the paragraph which are the predictor variables included to logistic regression
- 160 terms that were part of your set-up and hypothesis should not be removed from the model, even if non-significant (this is the statistic answer to your question!). You wanted to test the importance of the Light intensity. It was not important but this is a result! Keep in in your

models and discuss it in your discussion! (alternatively: show in the methods that it is not important, and shift the focus of the paper in hypotheses and analysis)

- Do not name one condition of the light treatment "light treatment", otherwise the reader cannot distinguish factor name and factor levels.
- Figure 2: state purpose of this set up (to attract glow worms). Figures with their legends should be extractable and useful outside the paper
- Complicated explanations in Lines 185-190 illustrate the problem with the name of the factor "distance", a confusing term for the location. Here it codes for a two-factor level categorical factor: illuminated or neighbouring, dark location. Meanwhile "distance" implies an interval-scaled gradient (something that increases or decreases with distance, make sure R really used as it "as factor"). In the methods you had named the same factor "locality", in Figure 2 distances are given in meters, in Figure 3 two levels "near" and "far". Be consistent. You should probably coin a better term, e.g. inside/outside light cone, factor "cone", or something similar, so everybody can understand your setup and analysis.
- Table 2: also provide information on the importance of your random factors date and locality (R marginal and R conditional)
- Fig. 3: the survival analysis <for proportion females not yet glowing> is somehow expressed around the corner. How about analyzing <nrs of females glowing>, or analyzing the <latency until female glows> (should yield similar results).
- Fig. 4: for a comprehensive legend you must give more information

The discussion is very limited. I miss considerations of

- predation risk for moving females at different light conditions (see e.g. Eccard et al. 2018 *Ins Cons*, Risk uniform and risk heterogenous conditions (Eccard and Liesenhohann 2008))
 - whether or not, or what kind of light heterogeneity occurs in nature
 - short term (this experiment) vs long term responses to illumination,
 - background knowledge on female and larval glow worm mobility - affecting their ability to leave or avoid lit areas as suggested
 - Sensory considerations, which light conditions allow male insects to pick up the signal, how sensory channels are affected by different ALAN techniques. Results of the mate attraction experiment should be picked up in the discussion
 - Importance of glow intensity of the dummy females: this variable was not significant but this needs to be discussed!!
- Line 241; direct
- Authors contribution statement in the beginning and end of the draft (line 252) are contradictory. Consider contribution of AK

Referee: 2

Comments to the Author(s)

The authors present a study investigating the effects of direct light sources on mate attraction behaviors in the glow worm. They found that females under higher ALAN have decreased glowing onset or don't glow, that the females don't move to get away from the light pollution, and that males are less attracted to glowing females that are under direct light sources. The study is well done and could be published as is at a more specific journal, however, I feel that this study does not meet the high standards of novelty for Proc B as is currently written. However, I do think that the authors could add to this manuscript and make it very attractive to readers of Proc B.

As I was reading through the manuscript I was very curious as to what the difference in achromatic contrast was between the female glow and the ambient lighting. This would not be hard to measure and then model using a non-specific visual system and it would greatly add to the story. I suggest measuring the radiance of the glow worm lures at the angle that a male would be seeing them and also measuring the radiance of the background substrate (street, grass, etc?) that a male would be viewing the glow against. Do this for each of the different distances of the glow worm to see what the contrast is. This would predict if males can even see the glow of the

glow worm under the different light treatments. If they easily can, then one could discuss how it isn't a visual detection issue but instead is likely a photophobic situation for males. I am very excited to see what you find.

Another avenue you could take it to set this paper apart from the previous work on Lampyridae, would be to look into the mechanism of the visual abilities of the female. What are they using to detect light levels to determine glowing or not? Are they using compound eyes or ocelli? If they are using only ocelli then it makes sense that they would not be able to determine where to move to for better lighting conditions. If they are using compound eyes, then what is their sensitivity to light levels and what is their acuity? Would they be able to see that there is an actual light gradient? I believe the authors are giving these beetles a lot of visual credit for being able to detect a light gradient of only one order of magnitude. This would be a very informative angle as well.

Lastly, and I would not suggest this as this would be the most time consuming, the authors could investigate actual fitness of these glowworms looking at mating rates and/or offspring production.

I hope my above suggestions are not overly daunting, but I do believe that this paper could take behavioral studies of light pollution to the next level and it is needed now in the field.

One minor detail that needs to be addressed - you use lux for ambient lighting but you use uW for radiance of glow worm lures. You need to either stick with photometric units (lux and lumens) or radiometric units (uWs or better yet, photons). This is very important.

Referee: 3

Comments to the Author(s)

The authors examine the movement patterns of female glowworms exposed to light pollution in experimental arenas in a laboratory. They also examine how light pollution impacts the likelihood that a female glowworm will mate under artificial lights in the field. The combination of complimentary field and laboratory experiments is one of this paper's assets.

This paper focuses on a very specific study organism--- glowworms. Many readers may find these organisms to be inherently interesting (insects that glow!). The addition of a few paragraphs in the introduction and discussion could give readers a broader perspective on how this study fits in with other research on how anthropogenic disturbances such as light pollution impact animal mating, movement, and courtship behaviors.

As I read through the paper, there were a few instances where I found myself wishing for more information. A few terms, such as "capital breeder" and "lantern," should be defined within the text for the benefit of readers who are unfamiliar with this study system. Further, the addition of more information about glowworm biology and natural history could help readers contextualize and interpret the results of this study. For example, readers may wish to know how far female glowworms typically travel at night, if this information is known.

Additional comments:

Line 30,68- The term "capital breeders" is used throughout this manuscript (e.g., lines 30, 68), but is not defined. This term may be unfamiliar to readers.

Line 37- Spell out "advertisements" instead of "adverts."

Line 89- What does the character immediately preceding "8 cm" in "(Ø 8 cm)" mean? Does this refer to one of the dimensions of the vials?

Line 93- Are light emission spectra available for the LED lights used in this experiment?

Line 95- Interesting! I'm trying to picture what the arenas might have looked like. Were the females most immediately in contact with the soil with the cardboard?

Line 99- Capitalize "pm."

Line 101- I'd like to know more about how the position of the female was recorded. Was the females' location marked within the area? Was the distance from the last known location measured with a ruler?

Line 103- "If a female settled multiple times, we used the last one as the settling location and time." How was "settling" defined? Additionally, I wonder if this approach would tend to underestimate the total distance moved by the female? For example, if a female moved 2 cm to the left, then 3 cm to the right along a straight line, would that be recorded as 5 cm or 1 cm traveled?

Line 105- Please define "lantern." Additionally, are there published data on the relationship between lantern size and female size? If so, please include a citation immediately after this statement: "To investigate if female size, which roughly correlates with lantern size and glow intensity,"

Line 123- Citation for this statistical analysis.

Line 148- How long do males stay with females after initial contact? Is it possible that some males might have left females prior to 1-2 AM, or is it more likely that males would stay with females all night?

Line 148- Capitalize AM.

Lines 205, 209- What is known about the distances female glowworms typically travel in the field at night? How does 1.2 m relate to the distances females typically travel?

Author's Response to Decision Letter for (RSPB-2019-2682.R0)

See Appendix A.

RSPB-2020-0806.R0

Review form: Reviewer 2

Recommendation

Accept with minor revision (please list in comments)

Scientific importance: Is the manuscript an original and important contribution to its field?

Excellent

General interest: Is the paper of sufficient general interest?

Marginal

Quality of the paper: Is the overall quality of the paper suitable?

Excellent

Is the length of the paper justified?

Yes

Should the paper be seen by a specialist statistical reviewer?

No

Do you have any concerns about statistical analyses in this paper? If so, please specify them explicitly in your report.

No

It is a condition of publication that authors make their supporting data, code and materials available - either as supplementary material or hosted in an external repository. Please rate, if applicable, the supporting data on the following criteria.

Is it accessible?

Yes

Is it clear?

Yes

Is it adequate?

Yes

Do you have any ethical concerns with this paper?

No

Comments to the Author

I understand that the authors don't want to add more analyses to the current study. I do hope that their future work does address my suggestions and questions.

I do have one more question: on lines 157-158 the authors claim that 1 to 2 lux is close to natural light levels at night. That is not true, that is not remotely true. Perhaps under a full harvest moon with good albedo (snow covered for instance), it may get to be between 1 and 10 lux. Most full moon conditions are between .1 and 1 lux. New moon conditions (which is more than 50 percent of the night) is less than .001 lux. So what do you mean? You even later say on line 248 that moonlight is around .05 to .1 lx, and again this actually ranges from .001 to 1 lux depending on lunar phase as well as lunar altitude and environmental conditions. So please rewrite lines 157 to 158 to be accurate and consistent with your discussion. I also think you need to caveat that your study's gradient is still quite bright.

Review form: Reviewer 4 (Sara Lewis)

Recommendation

Accept with minor revision (please list in comments)

Scientific importance: Is the manuscript an original and important contribution to its field?

Excellent

General interest: Is the paper of sufficient general interest?

Excellent

Quality of the paper: Is the overall quality of the paper suitable?

Excellent

Is the length of the paper justified?

Yes

Should the paper be seen by a specialist statistical reviewer?

No

Do you have any concerns about statistical analyses in this paper? If so, please specify them explicitly in your report.

No

It is a condition of publication that authors make their supporting data, code and materials available - either as supplementary material or hosted in an external repository. Please rate, if applicable, the supporting data on the following criteria.

Is it accessible?

Yes

Is it clear?

Yes

Is it adequate?

Yes

Do you have any ethical concerns with this paper?

No

Comments to the Author

Please see attached comments (Tyler 2013 also attached)

Decision letter (RSPB-2020-0806.R0)

18-May-2020

Dear Mrs Elgert:

Your manuscript has now been peer reviewed and the reviews have been assessed by an Associate Editor. The reviewers' comments (not including confidential comments to the Editor) and the comments from the Associate Editor are included at the end of this email for your reference. As you will see, the reviewers and the Editors have raised some concerns with your manuscript and we would like to invite you to revise your manuscript to address them.

Research ethics:

Use of animals and field studies:

All supplementary materials accompanying an accepted article will be treated as in their final form. They will be published alongside the paper on the journal website and posted on the online

figshare repository. Files on figshare will be made available approximately one week before the accompanying article so that the supplementary material can be attributed a unique DOI. Please try to submit all supplementary material as a single file.

Please submit a copy of your revised paper within three weeks. If we do not hear from you within this time your manuscript will be rejected. If you are unable to meet this deadline please let us know as soon as possible, as we may be able to grant a short extension.

Best wishes,
Professor Hans Heesterbeek
mailto:proceedingsb@royalsociety.org

Associate Editor
Comments to Author:

Dear Authors,

Thank you for the resubmission of your manuscript. I find it to be improved. The updated manuscript has now been seen by two established researchers in the field (one old and one new reviewer). Both reviewers see much merit in the manuscript and its findings, but have concerns. Notably, Reviewer 1 is concerned with the lux ratings described in the paper and also with the fact that many of their suggestions were not addressed in the revised document. In fact, many instances in the Response to Reviewer comments letter the authors used the excuse of the Proceedings B page limit to avoid addressing critical reviewer concerns, such as better detailing the disturbing effects of light pollution on organisms other than glow-worms. This is not acceptable. Yes, there are limits to what you can write given space concerns, but the important issues should still be addressed rather than ignored. Reviewer 2 has some concerns about terminology and other small comments.

In addition to the Reviewer concerns I have some of my own to this revised version of the manuscript. First, I think the abstract still needs work to better highlight the broader implications of the study for the readers of Proc B, and to highlight the specific novelty of this study. Second, unlike the reviewers of this round I still find that the introduction is a hard read. As said previously, a better delve into past literature on the effect of light pollution across a broader range of organisms is relatively inexistent and could be much better addressed. Please do so. Now the Introduction just goes straight into the glow-worm system, disregarding a more detailed and broad review of the literature. Third, I find that this paper as written, however lovely the experiment, is too specific for a general audience, and does not need to be this way given the relative lack of clear studies showing the behavioral effect of light pollution in nature. I only see broader implications in the very last paragraph. This needs to be better addressed. The manuscript is also still slightly choppy in places and the authors could benefit by having it re-read by colleagues outside their subject, before they turn in another revised version here.

Smaller concerns: Line 41 The sentence needs reworking. Line 47-50 Sentence grammatically incorrect. Also, if this sentence is supposed to be added to highlight the novelty of the current study then it is not clear that this is what you do here without expanding it. Line 265 remove 's' from movements

Again we thank the authors for their manuscript, and we look forward to a revised paper that addresses all the current concerns.

Reviewer(s)' Comments to Author:

Referee: 2

Comments to the Author(s).

I understand that the authors don't want to add more analyses to the current study. I do hope that their future work does address my suggestions and questions.

I do have one more question: on lines 157-158 the authors claim that 1 to 2 lux is close to natural light levels at night. That is not true, that is not remotely true. Perhaps under a full harvest moon with good albedo (snow covered for instance), it may get to be between 1 and 10 lux. Most full moon conditions are between .1 and 1 lux. New moon conditions (which is more than 50 percent of the night) is less than .001 lux. So what do you mean? You even later say on line 248 that moonlight is around .05 to .1 lx, and again this actually ranges from .001 to 1 lux depending on lunar phase as well as lunar altitude and environmental conditions. So please rewrite lines 157 to 158 to be accurate and consistent with your discussion. I also think you need to caveat that your study's gradient is still quite bright.

Referee: 4

Comments to the Author(s).

Please see attached comments (Tyler 2013 also attached)

Author's Response to Decision Letter for (RSPB-2020-0806.R0)

See Appendix B.

Decision letter (RSPB-2020-0806.R1)

22-Jun-2020

Dear Mrs Elgert

I am pleased to inform you that your manuscript entitled "Reproduction under light pollution: maladaptive response to spatial variation in artificial light in a glow-worm" has been accepted for publication in Proceedings B.

Open Access

Your article has been estimated as being 9 pages long. Our Production Office will be able to confirm the exact length at proof stage.

Paper charges

Sincerely,

Professor Hans Heesterbeek

Associate Editor:

Board Member

Comments to Author:

Dear Authors,

We wish to thank you for addressing all concerns and comments as best you could in this revision. The manuscript now reads really well and the focus much clearer in the introduction and discussion. I have no doubt that this paper will be well received by our readers and commend the authors for their attention to detail.

Appendix A

Authors' Response to Decision Letter for RSPB-2019-2682

Dear Prof. Hans Heesterbeek,

We are pleased to submit a revised version of our manuscript RSPB-2019-2682 for your consideration. Below is a detailed point-by-point response of how we have dealt with each of the reviewers' comments. The input provided by the reviewers has greatly strengthened the manuscript and we hope you will find the revision satisfactory. Of course, we are happy to make further changes as necessary. We look forward to a decision in due course.

We take the opportunity to thank the reviewers for their valuable input.

Kind regards,

Christina Elgert, Juhani Hopkins, Arja Kaitala and Ulrika Candolin

Associate Editor

Board Member: 1

Dear Authors,

Your manuscript has now been assessed by three excellent reviewers in the field. Although all reviewers see value in the topic they have valid and substantial concerns with the current version of the manuscript. These need to be specifically addressed if this paper is to compete for space in Proceedings B.

Namely, two reviewers mention issues with the language, description, and scope of the paper as written. This could be part of the reason why the strength in experimental design or findings of the paper are getting lost behind what could be vast improved writing.

Additionally, given that Proceedings is a general topic journal the authors must bring their relevance and findings more clearly into their broader implications in both the Abstract and Discussion; at least better than is done here.

It is my hope that these comments will help the authors in revising their manuscript for publication here. However, if the authors cannot address these issues and the others raised, I think this manuscript would be best served somewhere else.

RESPONSE: We now highlight the general importance of our work, to increase the interest to the general reader of Proceedings B. We do this both in the abstract (sentences added to the beginning and end), first paragraph of the introduction, and two new paragraphs at the end of the discussion.

All authors have checked the language, including Hopkins who is a native English speaker, and we hope it is now acceptable.

We have inserted more descriptions where possible. The page limit of Proceedings B restricts us from incorporating all suggestions by the reviewers. Yet, we hope we have found a good balance between the amount of details given and the length of the paper.

Reviewer(s)' Comments to Author:

Referee: 1

Review for RSPB-2019-2682, Jana Eccard, Animal Ecology, University of Potsdam

Paper: Reproduction under light pollution: maladaptive response to spatial variation in artificial light in a glow-

worm

GENERAL COMMENTS:

I love the experiment and the story, however I find the paper is not written very professional. Reading and commenting it felt like reading a student assignment, of a very good student but inexperienced paper writer (e.g. consistency in wording, finding functional terminology for the treatments, the function of a figure legend, the scope of the discussion). To improve this would have rather been the job of the senior authors before submission, not the job of a Proceedings reviewer. Still I kept reading and commenting below, since the experiments on glow worms are beautiful and light pollution is an important topic on the context of insect decline.

RESPONSE: We are very sorry to hear that the paper was not yet in submittable condition, and apologize for this. We hope the reviewer will find the revision to be of higher quality.

SPECIFIC COMMENTS:

Abstract

Line 28: i do not understand this effect when reading the abstract only, please rephrase

RESPONSE: We deleted the sentence as not essential, and to free space for discussing the general importance of our work.

Introduction:

Line 36 – 38: I would exchange line 36/37 with line 38 and start the article with the adaptations of organisms to natural light cycles

RESPONSE: Lines rearranged, lines 37- 48

The intro on disturbing effects of ALAN could be extended to other signaling systems of animals with luminescence, e.g. in the sea

RESPONSE: The page limit of Proceedings B forces us to limit ourselves to the most important issues. We have therefore decided not to enter into a detailed discussion about other organisms, but keep the discussion at a general level.

Line 70 plural: lights

RESPONSE: Corrected to nights, line 67

Methods:

Add information on scale and setup and purpose to the figure legend fig. 1

RESPONSE: We have added information to the legend of figure 1

Line 105: wether, not if

RESPONSE: changed to whether, line 114

Line 105 body size

RESPONSE: Sentence reformulated to “To investigate whether glow intensity influences responses to artificial light, we measured the maximum width of the pronotum (the structure that covers the dorsal surface of the thorax) of the female, as this correlates with lantern size, which roughly correlates with glow intensity” , lines 121 - 123

Line 107 the width of the pronotum of the female (the structure that covers the dorsal surface of the thorax) at its widest part – change to ... the maximum width of the pronotum (the structure that covers the dorsal surface of the thorax) of the female

RESPONSE: changed as suggested, lines 121 - 123

107 measured it for what? ... as a proxy of female body size (19)

RESPONSE: Restructured the sentence to be more informative: “To investigate whether glow intensity influences responses to artificial light, we measured the maximum width of the pronotum (the structure that covers the dorsal surface of the thorax) of the female, as this correlates with lantern size, which roughly correlates with glow intensity” , lines 121 - 123

Would latency to start glowing have been a variable to analyse?

RESPONSE: Latency to start glowing was the response variable of the survival analysis. We are sorry for the inconsistent use of terms and have now made it more consistent (replaced terms like ‘started to glow’, ‘time elapsed’, and “delayed onset” to “ latency”), to prevent further misunderstandings.

108 why are treatment and control so differently in numbers? Give a good reason since this is suspicious

RESPONSE: The difference in numbers is due to the control being used also for another experiment, which needed more replicates, carried out by our colleague Anna-Maria Borshagovski (unpublished material). This does in no way reduce its function as a control in our analysis, lines 124 - 125

109-116 state clearly at the start of the paragraph which are the predictor variables included to logistic regression

RESPONSE: Clarified at the start of the paragraph: “To analyse the impact of the presence of artificial light and the glow intensity of the female (measured as pronotum size) on behaviours that were binary, we used logistic regression,...” , lines 126 - 128

160 terms that were part of your set-up and hypothesis should not be removed from the model, even if non-significant (this is the statistic answer to your question!). You wanted to test the importance of the Light intensity. It was not important but this is a result! Keep in in your models and discuss it in your discussion! (alternatively: show in the methods that it is not important, and shift the focus of the paper in hypotheses and

analysis)

RESPONSE: Variables that do not influence the response variables and do not contribute to the model outcome can be removed from the final model when the reasons are clearly explained, which we do on lines 179 - 181. We are not in favour of altering hypotheses after an experiment has been carried out.

Do not name one condition of the light treatment "light treatment", otherwise the reader cannot distinguish factor name and factor levels.

RESPONSE: We suspect the reviewer is referring to the three glow intensities (low, medium, high) within the light treatment. This has now been clarified, lines 169 - 172.

Figure 2: state purpose of this set up (to attract glow worms). Figures with their legends should be extractable and useful outside the paper

RESPONSE: We have added the requested information to the legend of figure 2.

Complicated explanations in Lines 185-190 illustrate the problem with the name of the factor "distance", a confusing term for the location. Here it codes for a two-factor level categorical factor: illuminated or neighbouring, dark location. Meanwhile "distance" implies an interval- scaled gradient (something that increases or decreases with distance, make sure R really used as it "as factor"). In the methods you had named the same factor "locality", in Figure 2 distances are given in meters, in Figure 3 two levels "near" and "far". Be consistent. You should probably coin a better term, e.g. inside/outside light cone, factor "cone", or something similar, so everybody can understand your setup and analysis.

RESPONSE: We agree that the choice of the term 'distance' was less suitable. We have changed the term as suggested to 'within or outside the cone of light'. We thank the reviewer for pointing out the problem with the use of these terms.

Table 2: also provide information on the importance of your random factors date and locality (R marginal and R conditional)

RESPONSE: We assume the reviewer is referring to r^2 marginal (fixed factors only) and r^2 conditional (fixed + random factors). However, definitions for r-square are problematic in models with multiple error terms, and there is not yet a consensus on how to calculate them - that is why we are not reporting them. They are not the same 'variance explained' as in linear models. However, our data will be freely available so anybody interested can calculate them if needed.

Fig. 3: the survival analysis <for proportion females not yet glowing> is somehow expressed around the corner. How about analyzing <nrs of females glowing>, or analyzing the <latency until female glows> (should yield similar results).

RESPONSE: Our inconsistent use of terms had made it difficult to understand our analysis. We are now more consistent and use only latency to glow (not 'starting to glow', or 'time elapsed' etc). It is latency that we are analysing in the survival analysis.

Fig. 4: for a comprehensive legend you must give more information

RESPONSE: More information has been added, lines 222 - 224.

Discussion

The discussion is very limited.

RESPONSE: Because of the page limit of Proceedings B, we have to limit what we discuss. We have decided to not go in too many directions and keep speculations at a minimum, to avoid the paper becoming fragmented and too speculative. Thus, we have been selective in what we incorporate. Yet, we hope we have found a good balance in what we include in relation to the length of the paper.

I miss considerations of predation risk for moving females at different light conditions (see e.g. Eccard et al. 2018 Ins Cons, Risk uniform and risk heterogenous conditions (Eccard and Liesenhohann 2008)

RESPONSE: No information exists for glow-worms and we do not quite see how this would fit into the story. Predation risk might decrease with artificial light, which could be a good thing, but whether this is the case is unknown. It would probably not offset the cost of reduced mating attraction. However, as long as no information exists, we would rather not speculate on the topic.

whether or not, or what kind of light heterogeneity occurs in nature

RESPONSE: Discussed on lines 232 – 235 “small-scale spatial variation in light conditions is common in nature, not only in urban areas, but also along roads and in rural housing areas...” and 245 - 251: “The lack of appropriate responses to artificial light at night is probably a consequence of the natural absence of bright light at night...”

short term (this experiment) vs long term responses to illumination,

RESPONSE: We touch on this in our discussion of possible plastic and evolutionary responses to artificial light, lines 260 - 266. We would rather not dwell more on this until more information has been gained.

background knowledge on female and larval glow worm mobility - affecting their ability to leave or avoid lit areas as suggested

RESPONSE: We have added information regarding the movements of both adult females and larvae on lines 237 – 244, 250 – 251, 252 – 257.

Sensory considerations, which light conditions allow male insects to pick up the signal, how sensory channels are affected by different ALAN techniques. Results of the mate attraction experiment should be picked up in the discussion

RESPONSE: This is a big topic, which could fill a whole paper. We are currently focussing on part of these questions (current experiments) and before we have reliable results, we would rather not speculate on the matter (but stay tuned for coming papers).

Importance of glow intensity of the dummy females: this variable was not significant but this needs to be discussed!!

RESPONSE: This is discussed on lines 267 – 273: “Interestingly, the response of females to artificial light did not depend on their supposed glow intensity...”

Line 241; direct

RESPONSE: The section has been rewritten

Authors contribution statement in the beginning and end of the draft (line 252) are contradictory. Consider contribution of AK

RESPONSE: We apologize for the contradiction. AK contributed to the design of the experiment.

Referee: 2

The authors present a study investigating the effects of direct light sources on mate attraction behaviors in the glow worm. They found that females under higher ALAN have decreased glowing onset or don't glow, that the females don't move to get away from the light pollution, and that males are less attracted to glowing females that are under direct light sources. The study is well done and could be published as is at a more specific journal, however, I feel that this study does not meet the high standards of novelty for Proc B as is currently written. However, I do think that the authors could add to this manuscript and make it very attractive to readers of Proc B.

As I was reading through the manuscript I was very curious as to what the difference in achromatic contrast was between the female glow and the ambient lighting. This would not be hard to measure and then model using a non-specific visual system and it would greatly add to the story. I suggest measuring the radiance of the glow worm lures at the angle that a male would be seeing them and also measuring the radiance of the background substrate (street, grass, etc?) that a male would be viewing the glow against. Do this for each of the different distances of the glow worm to see what the contrast is. This would predict if males can even see the glow of the glow worm under the different light treatments. If they easily can, then one could discuss how it isn't a visual detection issue but instead is likely a photophobic situation for males. I am very excited to see what you find.

RESPONSE: These are very interesting ideas, and certainly worth doing, but we believe they require additional work and would be the topic of another paper. It would then be worth manipulating also the artificial light conditions and not only light or not light, as we do in the present experiments. We are currently planning such an experiment, where we manipulate the quality and spectrum of the artificial light, and we will discuss these issues in more detail in that paper – the page limit for Proceedings B papers would not allow us to include and discuss the results from the coming experiments in this paper.

Another avenue you could take it to set this paper apart from the previous work on Lampyridae, would be to look into the mechanism of the visual abilities of the female. What are they using to detect light levels to determine glowing or not? Are they using compound eyes or ocelli? If they are using only ocelli then it makes sense that they would not be able to determine where to move to for better lighting conditions. If they are

using compound eyes, then what is their sensitivity to light levels and what is their acuity? Would they be able to see that there is an actual light gradient? I believe the authors are giving these beetles a lot of visual credit for being able to detect a light gradient of only one order of magnitude. This would be a very informative angle as well.

RESPONSE: The females are known to be able to move in response to light, as they respond to green LED lights (lines 237 - 244). Moreover, even if the females would not be able to see the light gradient, they should still be able to discern between the bright and the dark end of the arena. Investigating the visual ability of females is another interesting topic, but goes beyond the purpose of this paper, which is ecologically oriented. It would be interesting to collaborate on the topic with researchers more familiar with the field, but it would be the topic of another paper.

Lastly, and I would not suggest this as this would be the most time consuming, the authors could investigate actual fitness of these glowworms looking at mating rates and/or offspring production.

RESPONSE: Yes, this would be really interesting to investigate, but as it turns out, raising glow-worms is not the easiest as it also demands the raising of snails, and the mortality commonly is high. Yet, this is included in future our plans.

I hope my above suggestions are not overly daunting, but I do believe that this paper could take behavioral studies of light pollution to the next level and it is needed now in the field.

One minor detail that needs to be addressed - you use lux for ambient lighting but you use uW for radiance of glow worm lures. You need to either stick with photometric units (lux and lumens) or radiometric units (uWs or better yet, photons). This is very important.

RESPONSE: Lux was chosen for the ambient light based on existing reference data for streetlights (lux is usually used in connection with light pollution) and our existing equipment. Unfortunately, our LED lures were too dim for accurate measurements with our lux-meter. However, we were able to measure the LED lures with a spectrophotometer and, thus, report their values in uW. The conversion from lux to uW for ambient light is unreliable, and we have therefore chosen not to do the conversion.

Referee: 3

Comments to the Author(s)

The authors examine the movement patterns of female glowworms exposed to light pollution in experimental arenas in a laboratory. They also examine how light pollution impacts the likelihood that a female glowworm will mate under artificial lights in the field. The combination of complimentary field and laboratory experiments is one of this paper's assets.

This paper focuses on a very specific study organism--- glowworms. Many readers may find these organisms to be inherently interesting (insects that glow!). The addition of a few paragraphs in the introduction and discussion could give readers a broader perspective on how this study fits in with other research on how anthropogenic disturbances such as light pollution impact animal mating, movement, and courtship behaviors.

RESPONSE: a few paragraphs have been added to increase the general interest of the paper to readers of Proceedings B, to the beginning of the introduction and the end of the discussion, and a few lines at the start

and the end of the abstract.

As I read through the paper, there were a few instances where I found myself wishing for more information. A few terms, such as “capital breeder” and “lantern,” should be defined within the text for the benefit of readers who are unfamiliar with this study system. Further, the addition of more information about glowworm biology and natural history could help readers contextualize and interpret the results of this study. For example, readers may wish to know how far female glowworms typically travel at night, if this information is known.

RESPONSE: We define lantern and capital breeder on lines 68 – 70 and 77 - 81, and added some other information on lines 67 - 83. We have added information regarding the movements of both adult females and larvae on lines 237 – 244, 250 – 251, 252 – 257.

Additional comments:

Line 30,68- The term “capital breeders” is used throughout this manuscript (e.g., lines 30, 68), but is not defined. This term may be unfamiliar to readers.

RESPONSE: Defined capital breeder on lines 68 - 70 “...glow-worms are capital breeders that depend on energy accumulated as larvae (they do not feed as adults)...”

Line 37- Spell out “advertisements” instead of “adverts.”

RESPONSE: Advertisements spelled out, line 39.

Line 89- What does the character immediately preceding “8 cm” in “(∅ 8 cm)” mean? Does this refer to one of the dimensions of the vials?

RESPONSE: Specifies “diameter”, which is now spelled out, line 99.

Line 93- Are light emission spectra available for the LED lights used in this experiment?

RESPONSE: Information added on lines 103 - 104.

Line 95- Interesting! I’m trying to picture what the arenas might have looked like. Were the females most immediately in contact with the soil with the cardboard?

RESPONSE: More information has been added to Figure 1, which pictures the arena. We now specify that the cardboard was 4 cm high, and ran along the middle of the arena, line 106.

Line 99- Capitalize “pm.”

RESPONSE: Capitalized to PM, lines 111 and 165

Line 101- I’d like to know more about how the position of the female was recorded. Was the females’ location marked within the area? Was the distance from the last known location measured with a ruler?

RESPONSE: we have added information that should clarify how the measurements were done, lines 113 - 121: “The borders of the arena had markings 10 cm apart, which we used to determine the position of the female and calculate the distance moved during each 20 min period.”

Line 103- “If a female settled multiple times, we used the last one as the settling location and time.” How was “settling” defined? Additionally, I wonder if this approach would tend to underestimate the total distance moved by the female? For example, if a female moved 2 cm to the left, then 3 cm to the right along a straight line, would that be recorded as 5 cm or 1 cm traveled?

RESPONSE: More information is given on how settling was estimated, lines 113 - 118 “We defined females as settling down when they had kept the same position over at least two consecutive observations, and recorded the location and time of settling. If a female settled multiple times, we used the last settling location and time.”

The reviewer is right in that our method probably underestimated the distance moved, but this is unlikely to influence the reliability of the results, lines 119 - 121 “This could underestimate the distance moved, as females may not move in a straight line, but the degree of underestimation is unlikely to differ across trials and treatments”

Line 105- Please define “lantern.” Additionally, are there published data on the relationship between lantern size and female size? If so, please include a citation immediately after this statement: “To investigate if female size, which roughly correlates with lantern size and glow intensity,”

RESPONSE: Defined lantern on lines 68 - 70, “The glow is produced in the lantern on the underside of the sixth and seventh segments of the abdomen, as well as in two spots on the eighth segment, through chemical reactions”, and added a citation “Hopkins J, Baudry G, Candolin U, Kaitala A. 2015 I'm sexy and I glow it: female ornamentation in a nocturnal capital breeder. *Biol. Lett.* 11, 20150599.”

Line 123- Citation for this statistical analysis.

RESPONSE: Inserted a citation, line 137 “Cox, David R (1972). “Regression Models and Life-Tables”. *Journal of the Royal Statistical Society, Series B.* 34 (2): 187–220.”

Line 148- How long do males stay with females after initial contact? Is it possible that some males might have left females prior to 1-2 AM, or is it more likely that males would stay with females all night?

RESPONSE: Males are highly unlikely to be able to escape from the dummy females, as they are funnel traps. Specified on lines 158 – 159, 168.

Line 148- Capitalize AM.

RESPONSE: Capitalized AM, line 167.

Lines 205, 209- What is known about the distances female glowworms typically travel in the field at night? How does 1.2 m relate to the distances females typically travel?

RESPONSE: We have added information regarding the movements of both adult females and larvae on lines 237 – 244, 250 – 251, 252 – 257.

Appendix B

i

Authors' Response to Decision Letter for RSPB-2020-0806

Dear Prof. Hans Heesterbeek,

We are pleased to submit a revised version of our manuscript RSPB-2020-0806 for your consideration. Below is a detailed point-by-point response of how we have attended to each of the reviewers' comments, and a copy of the manuscript with revisions made since the previous version marked as 'tracked changes'. The input provided by the reviewers has greatly strengthened the manuscript and we hope you will find the revision satisfactory. We look forward to a decision in due course.

We take the opportunity to thank the reviewers for their valuable input.

Kind regards,

Christina Elgert, Juhani Hopkins, Arja Kaitala and Ulrika Candolin

Associate Editor

Dear Authors,

Thank you for the resubmission of your manuscript. I find it to be improved. The updated manuscript has now been seen by two established researchers in the field (one old and one new reviewer). Both reviewers see much merit in the manuscript and its findings, but have concerns. Notably, Reviewer 1 is concerned with the lux ratings described in the paper and also with the fact that many of their suggestions were not addressed in the revised document. In fact, many instances in the Response to Reviewer comments letter the authors used the excuse of the Proceedings B page limit to avoid addressing critical reviewer concerns, such as better detailing the disturbing effects of light pollution on organisms other than glow-worms. This is not acceptable. Yes, there are limits to what you can write given space concerns, but the important issues should still be addressed rather than ignored. Reviewer 2 has some concerns about terminology and other small comments.

In addition to the Reviewer concerns I have some of my own to this revised version of the manuscript. First, I think the abstract still needs work to better highlight the broader implications of the study for the readers of Proc B, and to highlight the specific novelty of this study. Second, unlike the reviewers of this round I still find that the introduction is a hard read. As said previously, a better delve into past literature on the effect of light pollution across a broader range of organisms is relatively inexistent and could be much better addressed. Please do so. Now the Introduction just goes straight into the glow-worm system, disregarding a more detailed and broad review of the literature. Third, I find that this paper as written, however lovely the experiment, is too specific for a general audience, and does not need to be this way given the relative lack of clear studies showing the behavioral effect of light pollution in nature. I only see broader implications in the very last paragraph. This needs to be better addressed. The manuscript is also still slightly choppy in places and the authors could benefit by having it re-read by colleagues outside their subject, before they turn in another revised version here.

RESPONSE: First, We have altered the focus of the abstract to more strongly focus on maladaptive responses to anthropogenic environmental changes and the importance of unravelling the underlying mechanism, with our contribution revealing a behavioral mechanism behind the effect of artificial light, and offering support for the hypothesis that maladaptive responses are common.

Second, more information has been added to the introduction about effects of artificial light on other organisms (lines 48 – 53).

i

Third, the general interest of the paper has been increased by more strongly linking our work to the hypothesis that maladaptive responses are common when animals encounter conditions that they have not encountered in their recent evolutionary past (lines 43 – 47, 50 – 53). We have also extended the end part of the discussion by more broadly linking our results to the wider literature and stress the importance of investigating the behavioral mechanisms behind responses to disturbances. The text has in places been streamlined to be less choppy.

Smaller concerns:

Line 41 The sentence needs reworking.

RESPONSE: We have reworked the sentence: “Vast areas of the earth are currently lit up at night by direct light emission from various sources, such as streetlights, advertisements, buildings, and cars, and by indirect skyglow.” (lines 41 - 43).

Line 47-50 Sentence grammatically incorrect. Also, if this sentence is supposed to be added to highlight the novelty of the current study then it is not clear that this is what you do here without expanding it.

RESPONSE: The sentence has been removed and the topic discussed in the discussion: ‘This lack of knowledge hampers our ability to predict both short- and long-term effects of human disturbances on wildlife, as well as the development of effective management strategies to mitigate negative effects and requires more studies into the fitness consequences of behavioural responses to anthropogenic environmental changes.’ (lines 284 – 290)

Line 265 remove ‘s’ from movements

RESPONSE: We assume this refers to line 200 or 228 (no ‘movements’ on line 265). We have exchanged all ‘movements’ to ‘movement’ where suitable.

Again we thank the authors for their manuscript, and we look forward to a revised paper that addresses all the current concerns.

Reviewer(s)' Comments to Author:

Referee: 2

Comments to the Author(s).

I understand that the authors don't want to add more analyses to the current study. I do hope that their future work does address my suggestions and questions.

I do have one more question: on lines 157-158 the authors claim that 1 to 2 lux is close to natural light levels at night. That is not true, that is not remotely true. Perhaps under a full harvest moon with good albedo (snow covered for instance), it may get to be between 1 and 10 lux. Most full moon conditions are between .1 and 1 lux. New moon conditions (which is more than 50 percent of the night) is less than .001 lux. So what do you mean? You even later say on line 248 that moonlight is around .05 to .1 lx, and again this actually ranges from .001 to 1 lux depending on lunar phase as well as lunar altitude and environmental conditions. So please rewrite lines 157 to 158 to be accurate and consistent with your discussion. I also think you need to caveat that your study's gradient is still quite bright.

RESPONSE: Altered to: ‘...at 2.3 m from the pole, at 1–1.5 lx (peak intensity $\sim 0.004 \mu\text{W}/\text{cm}^2/\text{nm}$ at 455 nm), which is slightly brighter than natural light levels at night in the area in the summer (0.1–0.6 lx, peak intensity $0.0003\text{--}0.0016 \mu\text{W}/\text{cm}^2/\text{nm}$ at 460 nm, measured on May 31st and June 1st 2020 at 12.30 AM in an open area on both an overcast and a moonlit night).’, (line 162 – 165).

At our latitudes (N 59°51') the summer nights are rather bright, even without moonlight. Our experiment was carried out in the weeks around the summer solstice, at which point we have only about four hours of darkness per night. We had the opportunity to revisit the study area, and found environmental illumination to be approximately 0.1 – 0.6 lx (with lx-meter), with peak intensity $0.0003\text{--}0.0016 \mu\text{W}/\text{cm}^2/\text{nm}$ at 460 nm, measured on 31st May and 1st June 2020 at 12.30 AM in an open area during one overcast and one moonlit night (spectrophotometer and cosine corrector).

There seems to be some disagreement in the field regarding the brightness of moonlight. We have referenced Kyba, C., Mohar, A., & Posch, T. (2017). How bright is moonlight. *Astron. Geophys*, 58, 31-32 that states that in temperate latitudes, in the summer, typical moonlight lies between 0.05 to 0.1 lx, with a maximum of ~ 0.3 lx.

Referee: 4

Review of RSPB---2020---0806

This study examines the timely and important topic of how spatial variation in artificial light affects female glow behavior and male mate attraction in the glow---worm *Lampyrus noctiluca*.

This manuscript was a pleasure to read – it is well written, the experiments thoughtfully designed, the statistical analysis appropriate, and the results are presented clearly and concisely. Below we offer a few suggestions that we hope may further improve the paper.

Major suggestions:

1. Minorly confusing terminology: a) The variable called “Exposure to light” (i.e. in or out of the light cone) in the mate attraction experiment is a bit misleading because glowworms in the control treatment were also not exposed to light. Perhaps this variable could be called something like “position with respect to light fixture”?

RESPONSE: We agree and have changed the variable name to “position with respect to light fixture”, using the levels “inner position” and “outer position”.

b) Females’ pronotal width was measured as a proxy for their glow intensity, and based on previous work this seems reasonable (as explained in lines 120---123). However, the manuscript switches between these two variable names (lines 126 vs. 133). It seems good practice to use what was *actually* measured as the variable name, so perhaps this could be changed to “pronotum width (glow intensity)”.

RESPONSE: We have followed your advice and changed the variable name as suggested.

2. Additional citations: Although anecdotal, the attached 2013 paper by John Tyler (What do glow--- worms on their day off?) provides much relevant information on female glow---worm movement. Also relevant is

experimental work by Booth et al. (2004 J. Exp. Biol. 207, 2373---2378), which found that higher intensity lures were more attractive to *L. noctiluca* males.

RESPONSE: Thank you for these suggestions, we have added citations to these relevant sources (lines 262, 227 – 229).

3. Show data for male response to glow intensity: Although dummy females' glow intensity did not turn out to have a significant effect in the male attraction experiment, it seems important to show these results for both experimental treatments (in a supplementary figure if need be).

RESPONSE: Data from the 3 glow intensities has been added as a supplementary figure.

Minor suggestions:

Line 23 – Modify to read “A species *potentially* affected by light pollution...”?

RESPONSE: The sentence has been rewritten ‘ We investigated if females of the common glow-worm *Lampyrus noctiluca* – which glow in the night to attract males – mitigate negative effects of artificial light on mate attraction by adjusting the...’, lines 23 – 26.

Line 100 – Might be worth mentioning in the Introduction that glow---worm courtship is restricted to only 4 hours of darkness at your latitude – thus ALAN could have a more dramatic impact.

RESPONSE: Thank you, we have added the information to the method section, lines 107 – 109.

Line 103 --- perhaps this should read “[peak] intensity ~940 microWatts [per cm² per second]”?

RESPONSE: Thank you for pointing this out, we have exchanged “u” for “μ”. We used a spectrophotometer together with an integrating sphere for measuring the dummies, for the added measurements of ambient light we used both a lux-meter and a spectrophotometer together with cosine corrector. From the information on peak intensity (μW/nm, microwatts/nanometer) given by the spectrophotometer, we were able to calculate the total irradiance (μW). For clarity, we now have exchanged the total irradiance (μW) for the peak intensity (μW/nm) and added mention of the use of the integrating sphere. Lines 111 – 114.

Lines 161---62 --- this range of dummy female intensities may not have been great enough to see a difference (Steven’s power law)

RESPONSE: This seems unlikely, as the range was chosen to reflect natural variation at our study site and the difference in brightness was easy to see with human eye.

Line 171– It may be worth mentioning that small n (low statistical power) in the control treatment of the mate attraction experiment might have made it difficult to detect an effect of glow intensity

RESPONSE: This is now mentioned on lines 262 – 264 in the Discussion: ‘A larger sample size may be needed to detect the weaker effect of glow intensity on mate attraction as no significant effect of glow intensity on mate attraction was detected in the controls.’

Lines 212---14 – As mentioned above, it would be good to show results for the 3 glow intensities from both control and + light treatments.

RESPONSE: Results from the 3 glow intensities has been added as a supplementary figure.

Lines 255---257– add “potentially” to clarify that we don’t know if larvae actually do avoid light polluted areas

RESPONSE: The sentence has been rewritten: ‘Larvae have the potential to avoid light polluted areas by moving towards more undisturbed environments before they pupate to adults, as they can move up to 5 meters per hour [18], and do not pupate before they are 2–3 years old [18, 33, 42]. However, whether larvae can detect and respond to light pollution is currently unknown.’ Lines 241 – 246.

Lines 260---262– North American fireflies vary in emission wavelength ---“The degree to which this is an evolutionary change is unknown” — What does this mean? Clarify how this is relevant to adaptation to light pollution.

RESPONSE: We have clarified the sentence: ‘Fireflies of North-America differ in the wavelength of their glow depending on the local background, but the degree to which the differences have a genetic basis and indicate a potential for genetic adaptation is undetermined [44].’ (lines 248 – 251).

Lines 270---273 – Previous studies that have found an effect of glow intensity on male attraction should be cited again here. Also, the explanation given here does not seem consistent with the fact that glow intensity also did not influence male attraction in the dark controls (again, please show these data).

RESPONSE: Thank you for highlighting this. We have added citations to previous studies that have found an effect of brightness and an explanation for the lack of effect in the dark control: ‘ 
[revised manuscript text omitted]

~~natural light cycle at the latitude.~~ arkness is restricted to only 4 h during the height of the breeding
season at the latitude.

We investigated the responses of females to artificial light on the night after capture. We used a 100 cm
x 15 cm arena with a white light emitting diode, LED light (5 mm, cold white; peak intensity ~0.32
$\mu\text{W}/\text{nm}$ (microwatts/nanometer) ~~intensity ~940 μW , at 660 nm (red) with a secondary peak intensity of~~
~~~0.26 $\mu\text{W}/\text{nm}$ at 440 nm (blue), peak at 450 nm and 625 nm~~ as measured with spectrophotometer and
integrating sphere), at one of the short ends (Figure 1). Light intensity was 40 lx at the lit end, 1.5 lx in
the middle, and 0.5 lx at the dark end of the arena, measured with an AIRAM UVM-B lx-meter. We
covered the bottom of the arena with soil and placed a cardboard rectangle, 4 cm high, along the length
of the arena (in the middle of it) for the female to perch on, as females usually climb up on suitable
structures to enhance the visibility of the glow to flying males [28, 33]. A seashell was placed at each
short end of the arena for the female to hide under. We alternated the position of the LED light between
the two short ends among replicates.

We had two treatments: a light treatment and a control. We started each replicate at 11 PM by placing a
female on the cardboard in the middle of the arena and turning on the LED light in the light treatment
while leaving the LED light unlit in the control. We recorded the position of the female every 20 min
for 2 h and noted whether she had settled down and initiated glowing. The borders of the arena had
markings 10 cm apart, which we used to determine the position of the female and calculate the distance
moved during each 20 min period. We defined females as settled when they had kept the same position

over at least two consecutive observations and recorded the location and time of settling. If a female
settled multiple times, we used the last settling location and time in the analysis. We estimated distance
moved by summing distances moved during each 20 min period. This could underestimate the distance
moved, as females may not move in a straight line, however, this estimation was consistent across trials
and treatments. To investigate whether glow intensity influences responses to artificial light, we
measured the maximum width of the pronotum (the structure that covers the dorsal surface of the
thorax) of the female, as this correlates with lantern size, which roughly correlates with glow intensity
[29]. We tested 26 females in the light treatment and 63 females in the control.

To analyse the impact of the presence of artificial light and the pronotum width (glow intensity) of the
female_ ~~(measured as pronotum size)~~ on behaviours that were binary, we used logistic regression, with
separate models for each behaviour. The behaviours were: (i) whether a female started to glow, (ii) the
direction of movement in relation to the light source (towards or away), (iii) whether a female hid
under the shell or within the soil, and (iv) whether a female settled down.

To analyse the impact of artificial light on continuous measures, we used ANOVA, with separate
models for each measure. The measures were: (i) total distance moved during the 2 h of observation,
and (ii) the settling location (for those that settled). Pronotum widthsize (glow intensity) was included
as a covariate in the initial models, but as the interaction terms and the main effects were non-
significant, and the removal did not influence the significance of the other variables, it was removed in
the final model. To analyse the influence of artificial light on the latency to glowing, we used a time-to-
event analysis, Cox proportional hazards model [34]. The times were “right-censored” for females that
failed to glow by the end of the experiment. We checked that the data fulfilled the requirements of the
analyses. All analyses were performed using SPSS 25. Significance was designated as \$P < 0.05\$.

**Impact on mate attraction**

We conducted the experiment from June 6th to July 7th in 2017 in the surroundings of Tvärminne
 Zoological station. We selected six sites that lacked shadowing trees or bushes, and where we had
 detected glow-worms during earlier years. We manipulated the presence of artificial light by erecting a
 pole and attaching a white LED light (ANSI FL1 Standard: 35 lumen, beam distance 24 m) at the
 height of 1.7 m (Figure 2). The angle between the ground and the direction of the centre of the light
 was 55 degrees.

We placed two dummy females (LED lures, see construction in [29]) at two different distances, in a
 straight line from the pole (Figure 2). One dummy was placed in the inner position (within the cone of
 light from the pole (when lit), at 1.1 m from the pole, at 15–20 lx (peak intensity ~0.08 $\mu\text{W}/\text{cm}^2/\text{nm}$ at
 455 nm, measured with spectrophotometer and cosine corrector), which corresponds to light levels
 under common streetlights [2, 35]. The other dummy was placed ~~outside the cone of light~~ in the outer
 position, at 2.3 m from the pole, at 1–1.52 lx (peak intensity ~0.004 $\mu\text{W}/\text{cm}^2/\text{nm}$ at 455 nm), which is
 slightly brighter than natural light levels at night in the area in the summer (0.1–0.6 lx, peak intensity
 0.0003–0.0016 $\mu\text{W}/\text{cm}^2/\text{nm}$ at 460 nm, measured on May 31th and June 1st 2020 at 12.30 AM
[revised manuscript text omitted]
 effects of artificial light on populations is important, as it can ~~of the mechanisms~~
276 ~~underpinning responses to artificial light, and the consequences of these responses for the fitness of~~
277 ~~individuals and the dynamics of population, would~~ improve our ability to develop environment-friendly
lighting systems to minimize negative effects on wildlife [49]. The importance is increasing as
conditions are expected to worsen with the expansion of the human population and the increased use of
white LED lights [49, 50]. At a broader level, our study illuminates the importance of investigating
fitness consequences of behavioural responses to human-induced environment changes. An increasing
number of studies find organisms respond behaviourally to various human-caused disturbances, but
whether the responses are adaptive or not, and how they impact fitness components, are less known

[51, 52]. This lack of knowledge hampers our ability to ~~evaluate the~~ predict both short- and long-term
effects of human disturbances on ~~wildlife individuals and populations~~. Thus, we need to pay more
~~attention to the fitness consequences of behavioural responses if we are to elucidate the impact that~~
~~human activities have on population viability and, ultimately, on the composition of species~~
~~communities and the functioning of ecosystems~~. as well as the development of effective management
strategies to mitigate negative effects and requires more studies into the fitness consequences of
behavioural responses to anthropogenic environmental changes.

**Data accessibility**

The raw data is available in the Dryad database: <https://doi.org/10.5061/dryad.2z34tmphw>

**Competing interests**

The authors declare no competing interests.

**Authors' contributions**

CE, UC, AK and JH designed the research, CE performed the experiments, CE and UC analysed the
data, and CE and UC wrote the manuscript. All authors gave final approval for publication and agree to
be held accountable for the work performed therein.

**Acknowledgements**

We thank Anna-Maria Borshagovski and Gautier Baudry for assistance with the experiments, Topi
 Lehtonen and Lucy Katherine McLay for excellent comments on the manuscript, and Tvärminne
 Zoological Station for providing facilities.

**Funding**

The work was funded by Swedish Cultural Foundation in Finland (grant number 148370 to CE), Maj
 and Tor Nessling Foundation (grant number 202000239 to CE) and Academy of Finland (grant number
 294664 to AK).

**References**

[revised manuscript text omitted]

448